# Risk of cardiac events with azithromycin—A prediction model

**Haridarshan Patel** [1]☉*, **Robert J. DiDomenico**[1,2‡], **Katie J. Suda**[3‡], **Glen T. Schumock**[1‡], **Gregory S. Calip**[1,4‡], **Todd A. Lee**[1]☉

**1** Department of Pharmacy Systems, Outcomes, and Policy (PSOP), College of Pharmacy, University of Illinois at Chicago, Chicago, Illinois, United States of America, **2** Department of Pharmacy Practice, College of Pharmacy, University of Illinois at Chicago, Chicago, Illinois, United States of America, **3** Department of Medicine and Center for Health Equity Research and Promotions, VA Pittsburgh Healthcare System, University of Pittsburgh School of Medicine, Pittsburgh, Pennsylvania, United States of America, **4** Flatiron Health, Inc., New York, New York, United States of America

☉ These authors contributed equally to this work.
‡ These authors also contributed equally to this work.
* Haripatel86@gmail.com

**Data Availability Statement:** Data used in this analysis were obtained from and are the property of IBM MarketScan (US claims database). Any researcher requiring access to the raw data that were used to generate the analytical files can

## Abstract

Previous studies have suggested an increased risk of cardiac events with azithromycin, but the predictors of such events are unknown. We sought to develop and validate two prediction models to identify such predictors. We used data from Truven Marketscan Database (01/2009 to 06/2015). Using a split-sample approach, we developed two prediction models, which included baseline demographics, clinical conditions (Model 1), concurrent use of any drug (Model 1) and therapeutic class (Model 2) with a risk of QT-prolongation (CQT-Rx). Patients enrolled in a health plan for 365 days before and five days after dispensing of azithromycin (episodes). Cardiac events included syncope, palpitations, ventricular arrhythmias, cardiac arrest as a primary diagnosis for hospitalization including death. For each model, a backward elimination of predictors using logistic regression was applied to identify predictors in 100 random samples of the training cohort. Predictors prevalent in >50% of the models were included in the final model. A score for the Assessment of Cardiac Risk with Azithromycin (ACRA) was generated using the training cohort then tested in the validation cohort. A cohort of 20,134,659 episodes with 0.03% cardiac events were included. Over 60% included females with mean age of 40.1±21.3 years. Age, sex, history of syncope, cardiac dysrhythmias, non-specific chest pain, and presence of a CQT-Rx were included as predictors for Model-1 (c-statistic = 0.68). For Model-2 (c-statistic = 0.64), predictors included age, sex, anti-arrhythmic agents, anti-emetics, antidepressants, loop diuretics, and ACE inhibitors. ACRA score is available online (bit.ly/ACRA_2020). The ACRA score may help identify patients who are at higher risk of cardiac events following treatment with azithromycin. Providers should assess the risk-benefit of using azithromycin and consider alternative antibiotics among high-risk patients.

## Introduction

Drugs are a common cause of acquired Long QT syndrome [1–4]. Over 200 different drugs, including macrolides, have been associated with acquired Long QT syndrome [5]. Azithromycin,

access the data directly through IBM MarketScan under a license agreement, including the payment of appropriate license fees, between that third party and IBM MarketScan. Researchers interested in accessing the de-identified analytical data files for the above stated purposes should contact Hari Patel (hpatel64@uic.edu).

**Funding:** The author(s) received no specific funding for this work.

**Competing interests:** The authors have declared that no competing interests exist.

a macrolide, is commonly prescribed to treat acute infections and was considered to be the safest macrolide because of its low arrhythmogenic activity [6]. Because of its safety profile and a short duration of therapy, azithromycin is frequently prescribed (for every 1000 outpatient visits, 6.9 prescriptions were dispensed) [7]. However, in 2012, a study found an increased risk of cardiac deaths with the use of azithromycin within 5 days after therapy initiation. As a result, the U.S. Food and Drug Administration (FDA) issued a safety warning against the use of azithromycin among patients with known acquired Long QT syndrome, a history of torsades de pointes, congenital QT syndrome, bradyarrhythmias or uncompensated heart failure, and other proarrhythmic conditions such as hypokalemia, hypomagnesemia, and bradycardia [8]. The warning also cautioned the concurrent use of antiarrhythmic agents. The FDA warning included such risk factors, which were primarily identified for the development of acquired Long QT syndrome.

Despite the FDA warning, the profile of azithromycin users did not change markedly since 2012 [7]. In addition to the risk factors from the FDA warning, other factors may put patients receiving azithromycin therapy at a high risk of developing cardiac events. To date, no study has identified and confirmed the baseline predictors of cardiac events after azithromycin use. Our objective was to develop and internally validate two prediction models, which could be used by providers to identify characteristics that put patients at a high risk of developing cardiac events with azithromycin therapy. Both models differed by baseline predictors and aimed to be utilized by different providers. We also sought to create a risk score, which can be used with ease to identify patients at low, moderate, high and very high risk of developing cardiac events.

## Materials and methods

The predictor model was developed using data from the Truven Health Analytics MarkenScan Database [9], a private-sector health data resource, which reflects the healthcare experience of enrollees covered by the health benefit programs for employers. Data are collected from over 100 different health insurance companies and are nationally representative of commercially-insured enrollees and their dependents.

First, we identified patients who received one or more azithromycin prescription between January 1, 2009, and June 30, 2015. Multiple short-term dispensing of azithromycin therapy over time for a given patient were treated as episodes. Episodes were included if they had at least 365 days of continuous enrollment in a health plan before (baseline period), the dispense date of azithromycin (index date). Episodes were excluded if they had a missing enrollment ID, were exposed to azithromycin, clarithromycin, and erythromycin in the 30 days before and five days after the index date (follow-up period), or were prescribed azithromycin for more than 14 consecutive days (maintenance therapy).

Patient characteristics, including age, sex, geographic region, and insurance type, were collected at the index date. Clinical conditions were categorized using Clinical Classification Software (CCS) [10]. The software collapses diagnosis and procedure codes from the *International Classification of Diseases*, *9th Revision*, *Clinical Modification* (IDC-9-CM), which contains more than 14,000 diagnosis codes and 3,900 procedure codes. Over 300 chronic conditions from the level-three category were included as potential predictors.

Medications that prolong the QT interval or induce TdP were identified from a publicly available, well-established, and widely recognized list from the "CredibleMeds" website [5]. The drugs were categorized into 24 different therapeutic classes. Any QT-prolonging drugs prescribed within 90 days before the index date were considered. Concurrent use was defined based on an overlap of days' supply (with 15% allowed gap [11]) of QT-prolonging drugs with the index date of azithromycin prescription.

The outcome was measured as having a primary diagnosis during a hospitalization or first-listed emergency department (ED) visit based on ICD-9 codes for ventricular arrhythmia, palpitation, Long QT syndrome, cardiac arrest, syncope or death (ICD-9-CM codes: 426.82, 427, 427.1, 427.2, 427.4, 427.5, 427.9, 427.41, 427.42, 780.2, 785.1, 798, 798.1, 798.2, 798.9) within 5 days after the index date. This outcome definition was validated in previous studies with positive predictive values (PPVs) of over 80% [12,13]. We also applied an algorithm developed by another validation study to identify the outcomes originating from outpatient settings [14].

We developed two separate prediction models. Model 1 (for prescribers) included demographics, chronic comorbid conditions, and the presence of any concurrent QT-prolonging drug (as none, with at least one, and with one or more categories) as potential variables. Demographics and therapeutic classes of concurrent QT-prolonging drugs were included as predictors in Model 2 (for pharmacists). Models 1 and 2 included a total of 360 and 60 predictors, respectively.

An episode of azithromycin therapy was the unit of analysis. We used a split-sample design to derive and internally validate both prediction models. The total cohort was divided into two mutually exclusive populations for model development (training cohort = 66.6% of the sample) and internal validation (validation cohort = 33.4% of the sample). Both models were developed using the same training and validation cohorts. The training cohort was used for variable selection and model development, and the validation cohort was used to evaluate the performance of the final model. From the training cohort, a random sample containing 1,000,000 episodes of azithromycin therapy was selected. We then applied a backward elimination using logistic regression for variable selection in predictive modeling. With backward elimination, a criterion of $P$-value >0.15 was applied to remove variables from the model to generate a final model for the sample [15]. This process was repeated 100 times to determine the frequency of common predictors from 100 models. Predictors that were included in at least 50% of the models and were not correlated (with R-squared threshold of > 0.2) with other variables were selected for the final model (estimated using the entire training cohort). A sensitivity analysis was performed using both 75% and 100% as a predictor frequency for inclusion in the final model.

The statistical comparison between patients with and without a cardiac event included $t$-tests for the continuous measures and $\chi^2$ tests for the categorical variables. The validation cohort was used to assess discrimination and calibration of the final model [16]. Discrimination, the ability of the model to distinguish individuals who developed the outcome from those remained event free, was assessed using a concordance index (c-statistic). A receiver operating characteristic curve (ROC) was also generated. Calibration was assessed by plotting the observed versus predicted outcomes by decile of risk values with a Hosmer and Lemeshow test (test for goodness of fit).

We developed the Assessment of Cardiac Risk with Azithromycin (ACRA) scores to determine the probability of patients' risk of cardiac events with azithromycin therapy. The predicted probabilities, calculated from regression coefficient from the training cohort, were then used to assign a risk category in rank order (Low: <25%, Moderate: 25–50%, High: 51 to 75%, and Very High: >75%). The Brier score (range: 0 to 0.25, where a lower score indicates better performance) was used to evaluate the performance of risk categories using the validation cohort. All the statistical analyses were completed using SAS Enterprise Guide 7.1 (SAS Institute Inc., Cary, NC).

The use of data for this study was reviewed and determined to be exempt from human subjects research requiring informed consent by the Institutional Review Board of the University of Illinois at Chicago.

## Results

From January 1, 2009, to June 30, 2015, a total of 47,763,486 prescriptions of azithromycin were dispensed. Of these, 29,100,943 (60.9%) episodes were included based on the inclusion criteria of continuous enrollment before and after the index date. A total of 8,966,284 episodes (18.8%) were excluded for missing enrollment ID, exposure to a macrolide within 30 days before and five days after the index date, or if azithromycin was prescribed for a maintenance therapy (days supply >14 days). The final eligible cohort included 20,134,659 episodes of azithromycin therapy (12,172,949 unique patients). Almost 80% of the episodes included patients who were 18 years or older, 60% were female, and the majority (42.9%) from the Southern US (Table 1). There were 7,009 (0.03%) cardiac events identified among the included azithromycin

**Table 1. Patient characteristics.**

| Variables | Total | Without a Cardiac Event | With a Cardiac Event | Standardized Difference |
|---|---|---|---|---|
| | (N = 20,134,659) | (N = 20,127,650) | (N = 7,009) | |
| **Age Group (in years)** | | | | |
| Less than or equal to 17 | 4,348,886 (21.6) | 4,348,211 (21.6) | 675 (9.6) | 33.5 |
| 18 to 34 | 3,878,631 (19.3) | 3,877,375 (19.3) | 1,256 (17.9) | 3.6 |
| 35 to 44 | 3,230,809 (16.0) | 3,229,824 (16.0) | 985 (14.1) | 5.3 |
| 45 to 54 | 3,543,373 (17.6) | 3,542,155 (17.6) | 1,218 (17.4) | 0.5 |
| 55 to 64 | 3,375,314 (16.8) | 3,373,923 (16.8) | 1,391 (19.8) | 7.8 |
| 65 & older | 1,757,646 (8.7) | 1,756,162 (8.7) | 1,484 (21.2) | 35.6 |
| **Sex** | | | | |
| Male | 7,951,287 (39.5) | 7,948,404 (39.5) | 2,883 (41.1) | 3.3 |
| Female | 12,183,372 (60.5) | 12,179,246 (60.5) | 4,126 (58.9) | 3.3 |
| **Region** | | | | |
| Northeast Region | 3,063,323 (15.2) | 3,062,198 (15.2) | 1,125 (16.1) | 2.5 |
| North Central Region | 4,877,579 (24.2) | 4,875,728 (24.2) | 1,851 (26.4) | 5.1 |
| South Region | 8,645,336 (42.9) | 8,642,513 (42.9) | 2,823 (40.3) | 5.3 |
| West Region | 3,181,028 (15.8) | 3,179,929 (15.8) | 1,099 (15.7) | 0.3 |
| Unknown Region | 367,393 (1.8) | 367,282 (1.8) | 111 (1.6) | 1.5 |
| **Type of Insurance** | | | | |
| Preferred provider organization | 12,099,495 (60.1) | 12,095,602 (60.1) | 3,893 (55.5) | 9.3 |
| Health maintenance organization | 2,568,407 (12.8) | 2,567,522 (12.8) | 885 (12.6) | 0.6 |
| Point-of-service plan | 1,393,092 (6.9) | 1,392,624 (6.9) | 468 (6.7) | 0.8 |
| Consumer directed health plan | 1,280,502 (6.4) | 1,280,136 (6.4) | 366 (5.2) | 5.1 |
| High deductible health plan | 684,604 (3.4) | 684,393 (3.4) | 211 (3.0) | 2.3 |
| Exclusive provider organization | 278,014 (1.4) | 277,953 (1.4) | 61 (0.9) | 4.7 |
| Other | 115,111 (0.6) | 115,069 (0.6) | 42 (0.6) | 0 |
| Missing/Unknown | 525,919 (2.6) | 525,692 (2.6) | 227 (3.2) | 3.6 |
| **Year** | | | | |
| 2010 | 3,378,933 (16.8) | 3,377,870 (16.8) | 1,063 (15.2) | 4.4 |
| 2011 | 4,222,167 (21.0) | 4,220,666 (21.0) | 1,501 (21.4) | 1.0 |
| 2012 | 4,641,036 (23.0) | 4,639,459 (23.1) | 1,577 (22.5) | 1.4 |
| 2013 | 3,198,041 (15.9) | 3,196,828 (15.9) | 1,213 (17.3) | 3.8 |
| 2014 | 3,199,514 (15.9) | 3,198,385 (15.9) | 1,129 (16.1) | 0.5 |
| 2015 | 1,494,968 (7.4) | 1,494,442 (7.4) | 526 (7.5) | 0.4 |

Note: All comparisons between episodes with and without a cardiac event were statistically significant (p<0.05).

episodes. Syncope and palpitations (65.5% and 24.7%, respectively) were the most common type of cardiac event, and five patients had died during the follow-up period (Table 2).

Among episodes with and without a cardiac event, the differences in demographics were all statistically significant (P<0.05) due to large sample sizes (Table 1). Overall, the episodes with a cardiac event included a higher percentage of patients who were more than 65 years old and had comprehensive coverage. Episodes without a cardiac event had a higher percentage of patients who were less than 17 years old. Episodes including patients with a cardiac event had a higher prevalence of baseline cardiovascular disease including hypertension, heart failure, and coronary atherosclerosis (Table 3).

Among episodes with a cardiac event, 22.3% had at least one, and 10.4% had two or more concurrent QT-prolonging drugs. Among episodes with at least one or more concurrent QT-prolonging drugs, the most common class were antidepressants (31.9%), followed by opioid agonists (8.6%). We found no statistical differences in predictor variables between the training and validation cohorts.

For Model 1, the number of predictors in 10%, 25%, 50%, 75%, and 100% of the models from the training cohort were 151, 20, 6, 5, and 1, respectively. The final predictors (with frequency) included a history of syncope (100%), age (97%), cardiac dysrhythmias (91%), presence of QT-prolonging drug (81%), non-specific chest pain (77%), and sex (52%). The effect estimates of predictors including in Model 1, were consistent between both training and validation cohorts (Table 4). History of syncope had the highest odds of predicting a cardiac event. Age group greater than 65 years old had greater odds of a cardiac event compared to 35 to 44 years old patients.

Model 2 had 31, 16, 7, 2 and 1 number of predictors in 10%, 25%, 50%, 75%, and 100% of the models, respectively. The final predictors (with frequency) included age (100%), anti-arrhythmic agents (89%), sex (56%), anti-emetics (56%), antidepressants (55%), loop diuretics (53%), and ACE inhibitors (52%). In the validation cohort, the use of concurrent ACE inhibitors was not a significant predictor of cardiac events (Table 5). Similar to Model 1, older patients (> 65 years of age) predicted a high likelihood of a cardiac event. Episodes with concurrent use of an antiarrhythmic or antiemetic drug with azithromycin had the greatest odds of a cardiac event.

The Hosmer-Lemeshow goodness of fit test resulted in a good fit for both models (Model 1: P-values of 0.08 and 0.06 for the training and validation cohorts, respectively, Model 2: 0.99 and 0.97). Discrimination of the model was good: c-statistic was 0.68 and 0.64 for Model 1 and 2, respectively. The observed and predicted events based on the ACRA scores from both Models are included in Fig 1. Sensitivity analysis using the various thresholds (75% and 100%) of the frequency of predictor variables was performed (See Tables 4 and 5). At the threshold of

**Table 2. Frequency of the outcome of cardiac event.**

|  | Total (N = 20,134,659) |
|---|---|
| Outcome (at least one or more of the following conditions) | 7009 (0.03) |
| Syncope | 4599 (65.62) |
| Palpitations | 1733 (24.73) |
| Cardiac arrest | 353 (5.04) |
| Cardiac dysrhythmia | 273 (3.89) |
| Paroxysmal ventricular tachycardia | 114 (1.63) |
| Ventricular fibrillation | 36 (0.51) |
| Long QT syndrome | 17 (0.24) |
| Death | 5 (0.07) |
| Ventricular flutter | 2 (0.03) |

**Table 3. Comorbid conditions of cohort by outcome.**

|  | Total | Without a Cardiac Event | With a Cardiac Event | Standardized Difference |
|---|---|---|---|---|
|  | (N = 20,134,659) | (N = 20,127,650) | (N = 7,009) |  |
| Hypertension | 3,168,455 (15.7) | 3,166,417 (15.7) | 2,038 (29.1) | 32.6 |
| Cardiac arrest and ventricular fibrillation | 5,854 (0.0) | 5,839 (0.0) | 15 (0.2) | 6.3 |
| Congestive heart failure | 168,921 (0.8) | 168,616 (0.8) | 305 (4.4) | 22.8 |
| Acute myocardial infarction | 43,498 (0.2) | 43,427 (0.2) | 71 (1.0) | 10.4 |
| Coronary atherosclerosis | 642,146 (3.2) | 641,461 (3.2) | 685 (9.8) | 27 |
| Pulmonary heart disease | 73,738 (0.4) | 73,670 (0.4) | 68 (1.0) | 7.2 |
| Acute cerebrovascular disease | 86,091 (0.4) | 85,950 (0.4) | 141 (2.0) | 14.7 |
| Transient cerebral ischemia | 70,271 (0.3) | 70,173 (0.3) | 98 (1.4) | 12 |
| Peripheral and visceral atherosclerosis | 221,525 (1.1) | 221,306 (1.1) | 219 (3.1) | 14 |

Note: All comparisons between episodes with and without a cardiac event were statistically significant (p<0.05).

75%, sex was eliminated, and at 100%, only syncope remained as a significant predictor in Model 1. For Model 2, the threshold of 75% included age and antiarrhythmic agents, and at 100%, only age remained the significant predictor. The threshold of 50% was appropriate for developing the final models based on the model discrimination (c-statistic > 0.67) and the clinical relevance of the predictors.

## Discussion

Using data from a large claims database, we identified the predictors of a cardiac event among patients treated with azithromycin therapy in an outpatient setting. The occurrence of cardiac events among azithromycin users is low (0.03%), but patients with one or more predictors from our models might be at a high risk of such events. Using the ACRA calculator may help with the appropriate use of azithromycin in outpatient settings.

**Table 4. Logistic regression model of predictors of cardiac events among episodes of azithromycin therapy—Model 1.**

|  | Training Cohort | | | Validation Cohort | |
|---|---|---|---|---|---|
|  | Model A (50%) | Model B (75%) | Model C (100%) | Final Model | Beta-Coefficients[1] |
| **Syncope** | 3.90 (3.47–4.38) | 3.89 (3.47–4.37) | 7.19 (6.45–8.01) | 3.66 (3.11–4.31) | 1.30 |
| **Age groups (in years)** | | | | | |
| ≤17 vs 35 to 44 | 0.54 (0.48–0.61) | 0.55 (0.49–0.62) | - | 0.65 (0.55–0.77) | -0.43 |
| 18 to 34 vs 35 to 44 | 1.10 (1.00–1.22) | 1.10 (0.99–1.22) | - | 1.15 (0.99–1.33) | 0.14 |
| 45 to 54 vs 35 to 44 | 1.03 (0.93–1.15) | 1.04 (0.94–1.15) | - | 1.11 (0.95–1.28) | 0.10 |
| 55 to 64 vs 35 to 44 | 1.12 (1.01–1.23) | 1.12 (1.01–1.24) | - | 1.31 (1.13–1.51) | 0.27 |
| ≥65 vs 35 to 44 | 1.79 (1.61–1.98) | 1.81 (1.63–2.01) | - | 1.97 (1.70–2.28) | 0.68 |
| **Cardiac dysrhythmias** | 1.99 (1.82–2.18) | 2.00 (1.83–2.18) | - | 2.30 (2.04–2.60) | -0.43 |
| **Concurrent QT-prolonging drug** | | | | | |
| At least one vs None | 1.32 (1.22–1.42) | 1.30 (1.21–1.40) | - | 1.29 (1.17–1.44) | 0.83 |
| Two or more vs None | 1.91 (1.73–2.12) | 1.88 (1.70–2.08) | - | 1.74 (1.50–2.01) | 0.26 |
| **Non-specific chest pain** | 1.67 (1.54–1.82) | 1.67 (1.53–1.81) | - | 1.74 (1.55–1.96) | 0.55 |
| **Male vs Female** | 1.17 (1.11–1.25) | - | - | 1.15 (1.06–1.25) | 0.56 |

[1]Beta-coefficient intercept value was -8.40.

Model B and C include sensitivity analysis results (predictors present in 75% and 100% of 100 models with a c-statistic value of 0.6712 and 0.5320, respectively).

Predictors from Model A (50%) were selected for validation and final model development based on c-statistic value (0.6740).

**Table 5. Logistic regression model of predictors of cardiac events among episodes of azithromycin therapy—Model 2.**

| | Training Cohort | | | Validation Cohort | |
| --- | --- | --- | --- | --- | --- |
| | Model A (50%) | Model B (75%) | Model C (100%) | Final Model | Beta-Coefficients[1] |
| **Age group (in years)** | | | | | |
| ≤17 vs 35 to 44 | 0.49 (0.43–0.55) | 0.48 (0.43–0.54) | 0.48 (0.43–0.54) | 0.58 (0.49–0.69) | -0.54 |
| 18 to 34 vs 35 to 44 | 1.07 (0.97–1.19) | 1.05 (0.95–1.16) | 1.05 (0.95–1.16) | 1.10 (0.95–1.28) | 0.09 |
| 45 to 54 vs 35 to 44 | 1.08 (0.98–1.20) | 1.10 (0.99–1.22) | 1.10 (0.99–1.22) | 1.16 (1.02–1.35) | 0.15 |
| 55 to 64 vs 35 to 44 | 1.22 (1.11–1.35) | 1.27 (1.15–1.40) | 1.28 (1.16–1.42) | 1.44 (1.25–1.67) | 0.37 |
| 65 & older vs 35 to 44 | 2.35 (2.12–2.60) | 2.55 (2.31–2.81) | 2.67 (2.42–2.95) | 2.59 (2.24–3.00) | 0.95 |
| **Antiarrhythmic agents** | 3.78 (3.01–4.74) | 4.22 (3.37–5.28) | - | 4.32 (3.21–5.81) | 0.12 |
| **Male vs Female** | 1.16 (1.09–1.23) | - | - | 1.13 (1.04–1.23) | 1.46 |
| **Antiemetics[2]** | 2.75 (2.10–3.60) | - | - | 3.40 (2.41–4.80) | 1.22 |
| **Antidepressants[2]** | 1.34 (1.23–1.47) | - | - | 1.20 (1.06–1.37) | 0.19 |
| **Loop diuretics[2]** | 1.54 (1.33–1.77) | - | - | 1.68 (1.39–2.04) | 0.52 |
| **ACE inhibitors[2]** | 1.52 (1.28–1.80) | - | - | 1.23 (0.95–1.60) | 0.21 |

Model B and C include sensitivity analysis results (predictors present in 75% and 100% of 100 models with a c-statistic value of 0.6179 and 0.6162, respectively).

Predictors from Model A (50%) were selected for validation and final model development based on c-statistic value (0.6318).

[1]Beta-cofficient intercept value was -8.23.

[2]The following drugs were included within each therapeutic class: Antiarrhythmic Agent (Quinidine, Propafenone, Dronedarone, Flecainide, Amiodarone), Antiemetics (Dolasetron, Granisetron, Ondansetron, Metoclopramide), Antidepressants (Desvenlafaxine, Maprotiline, Desipramine, Fluvoxamine, Imipramine, Fluoxetine, Venlafaxine, Mirtazapine, Amitriptyline, Trazodone, Escitalopram, Citalopram, Sertraline), Loop Diuretics (Furosemide), and ACE Inhibitors (Captopril, Fosinopril, Moexipril, Quinapril, Enalapril, Benazepril, Lisinopril).

Previous literature has identified several risk factors of Long QT syndrome in a general population. The risk of cardiac event or death was strongly associated with increasing age, male sex, and presence of underlying cardiovascular disease, including heart failure [17–20]. Although no study has directly identified the predictors of cardiac events among azithromycin users, a few risk evaluation studies have examined subgroups based on the general understanding of the risk factors that lead to a cardiac event. In a 2012 study from Ray *et al*. [21] focused on evaluating the risk of cardiac events among azithromycin users, there was a substantially higher risk of deaths among patients with pre-existing cardiovascular disease. In another study by Choi *et al*. [22], the risk of QT prolongation after azithromycin use was evaluated using data from a university hospital in Korea. The investigators examined the effects of age and sex, but only found older age (>65 years) to be associated with a higher risk of cardiac events. These risk factors were mostly consistent with our findings. In contrast to previous literature, we identified that middle-age (55 to 64 years old) was also predictive of a cardiac event.

The FDA warning in 2012 was primarily based on an understanding of the risk factors of Long QT syndrome in a general population [8]. The risk factors included older age, males, history of heart disease, especially heart failure, hypokalemia, and bradycardia. While we were not able to ascertain the ongoing electrolyte abnormalities from our database, we confirmed all of the risk factors from our study, except for heart failure. From our model development exercise, congestive heart failure was present in only 26% of the model (out of 100 models) and was not a significant predictor of the outcome. It is possible that heart failure is a heterogeneous condition and certain phenotypes may be prone to cardiac arrhythmias. On the contrary, we found additional risk factors including a history of syncope and non-specific chest pain, to be predictive of a cardiac event. Syncope is a clinical consequence of many conditions including cardiac arrhythmias and non-specific chest pain has been previously associated with a sudden cardiac event [23–25] Among azithromycin users, these conditions may further increase the risk of a

A. Model 1

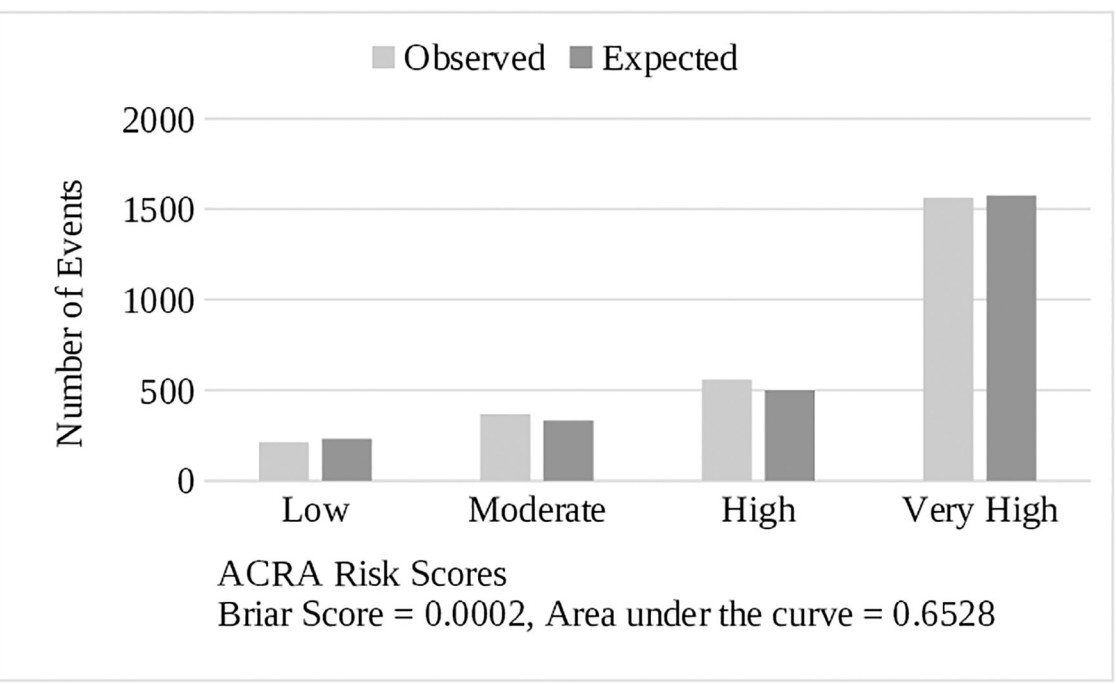

B. Model 2

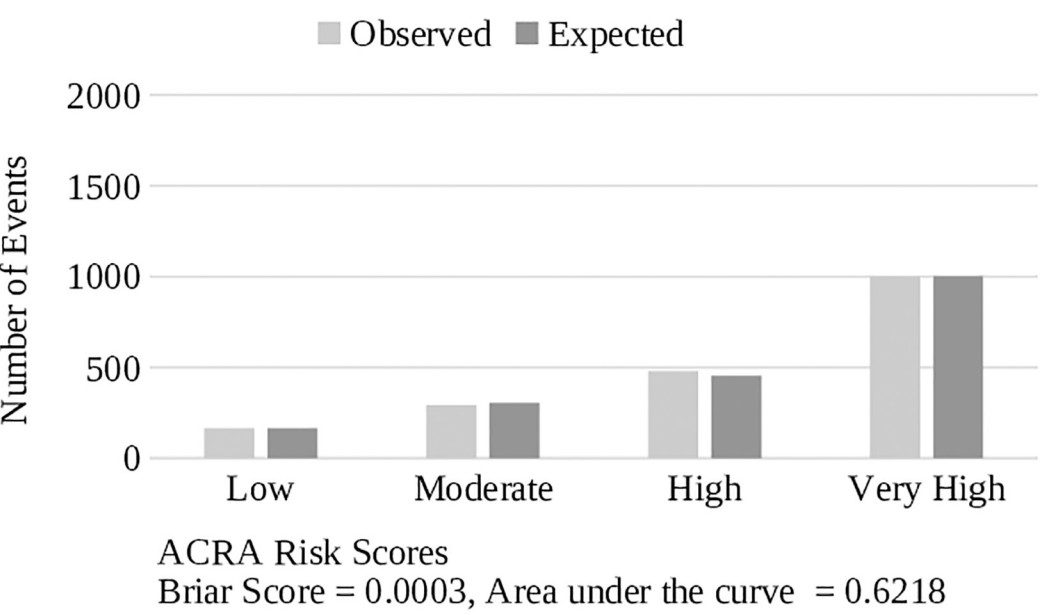

**Fig 1. Comparison of predicted and observed cardiac events from ACRA score.**

cardiac event. The FDA warning exclusively focused on antiarrhythmic agents as a risk factor. Previous studies that evaluated the risk of cardiac events with azithromycin did not examine which class of QT-prolonging drugs put patients at a higher risk. Our findings from Model 1

add to this knowledge, showing the risk with at least one or more QT-prolonging drugs was almost 30% higher and 75% higher with the use of two or more QT-prolonging drugs. In Model 2, we examined specific therapeutic classes of QT-prolonging drugs as predictors and found that in addition to antiarrhythmic agents, several other therapeutic classes including antiemetics, antidepressants, ACE inhibitors, and loop diuretics posed a similar risk of the outcome. ACE inhibitors have a protective effect on cardiac remodeling, but the potassium-sparing effects may help explain why it was one of the predictors of cardiac events with azithromycin [26].

We developed the two separate ACRA scores to be used at different inflection points in the healthcare system. Physicians are privy to patients' history of clinical conditions such as syncope or cardiac dysrhythmias, whereas community pharmacists are less likely to know patients baseline conditions but have more extensive information on prior and concurrent drug use. Prescribers may use the ACRA score from Model 1 to identify high-risk patients based on their baseline conditions before prescribing azithromycin. On the other hand, ACRA score from Model 2 can help pharmacists assess the appropriate use of concurrent QT-prolonging drugs with azithromycin.

Our findings must be interpreted in the context of several limitations. With a retrospective, observational study design, unmeasured confounding, and measurement error are always possible. To develop our prediction models, we used data from administrative health care claims, which are primarily used for billing purposes, not research. We were unable to examine essential risk factors including, race, smoking status, use of over-the-counter medications, electrolyte abnormalities, QT prolongation, and body mass index as these variables were not available in the database. Despite this limitation, we used an extensive list of predictors ($> 360$) in both of our prediction models. Certain clinical conditions, including the outcome, may be underreported, and most severe cases might have led to hospitalization. Although we were able to measure the outcome of death using the ICD-9 codes, we lacked information on death occurring in settings other than inpatient hospitalization. We included palpitation, a non-specific symptom, as one of the cardiac events and assumed that patients experienced such event due to treatment and not because of a baseline condition. We assumed the conditions measured in the year before azithromycin prescription would be most relevant and probably taken into consideration at the time of prescribing.

## Conclusions

In summary, we developed and internally validated two prediction models to predict the risk of a cardiac event among patients treated with azithromycin therapy. The ACRA score may help both general practitioners and pharmacists identify patients who are at considerable risk of a cardiac event after being exposed to azithromycin therapy. Further work is needed to identify factors not included in our study and validate the model in other populations.

## Author Contributions

**Conceptualization:** Haridarshan Patel, Robert J. DiDomenico, Katie J. Suda, Glen T. Schumock, Gregory S. Calip, Todd A. Lee.

**Data curation:** Haridarshan Patel, Katie J. Suda, Glen T. Schumock, Gregory S. Calip, Todd A. Lee.

**Formal analysis:** Haridarshan Patel, Katie J. Suda, Gregory S. Calip, Todd A. Lee.

**Funding acquisition:** Haridarshan Patel.

**Investigation:** Haridarshan Patel, Katie J. Suda, Glen T. Schumock, Gregory S. Calip, Todd A. Lee.

**Methodology:** Haridarshan Patel, Robert J. DiDomenico, Katie J. Suda, Glen T. Schumock, Gregory S. Calip, Todd A. Lee.

**Project administration:** Haridarshan Patel, Katie J. Suda, Glen T. Schumock, Gregory S. Calip, Todd A. Lee.

**Resources:** Haridarshan Patel, Robert J. DiDomenico, Katie J. Suda, Glen T. Schumock, Gregory S. Calip, Todd A. Lee.

**Software:** Haridarshan Patel, Todd A. Lee.

**Supervision:** Haridarshan Patel, Robert J. DiDomenico, Katie J. Suda, Glen T. Schumock, Gregory S. Calip, Todd A. Lee.

**Validation:** Haridarshan Patel, Gregory S. Calip, Todd A. Lee.

**Visualization:** Haridarshan Patel, Todd A. Lee.

**Writing – original draft:** Haridarshan Patel, Robert J. DiDomenico, Katie J. Suda, Glen T. Schumock, Gregory S. Calip, Todd A. Lee.

**Writing – review & editing:** Haridarshan Patel, Robert J. DiDomenico, Katie J. Suda, Glen T. Schumock, Gregory S. Calip, Todd A. Lee.

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
