## [Decision Letter · Decision Letter 0]

22 Jul 2020

PONE-D-20-15013

Risk of Cardiac Events with Azithromycin – A Prediction Model

PLOS ONE

Dear Dr. Patel,

Thank you for submitting your manuscript to PLOS ONE. After careful consideration, we feel that it has merit but does not fully meet PLOS ONE’s publication criteria as it currently stands. Therefore, we invite you to submit a revised version of the manuscript that addresses the points raised during the review process.

We look forward to receiving your revised manuscript.

Kind regards,

Antonio Cannatà

Academic Editor

PLOS ONE

Journal Requirements:

Reviewers' comments:

Reviewer's Responses to Questions

**Comments to the Author**

1. Is the manuscript technically sound, and do the data support the conclusions?

Reviewer #1: Yes

Reviewer #2: Yes

Reviewer #3: Yes

2. Has the statistical analysis been performed appropriately and rigorously? 

Reviewer #1: Yes

Reviewer #2: Yes

Reviewer #3: Yes

3. Have the authors made all data underlying the findings in their manuscript fully available?

Reviewer #1: No

Reviewer #2: Yes

Reviewer #3: Yes

4. Is the manuscript presented in an intelligible fashion and written in standard English?

Reviewer #1: Yes

Reviewer #2: Yes

Reviewer #3: Yes

5. Review Comments to the Author

Reviewer #1: Patel et al. in their paper explore the predictors of cardiac adverse event during therapy with azithromycine. Even though azithromycine has been found to be the less pro-arrhythmic macrolide, the arrhythmic risk prediction during therapy with azithromycine has not been extensively investigated previously. Therefore, a tool to predict this risk may have a relevant clinical application. The paper is well written, well conducted, and statistically accurate. The study is conducted on a large population and the number of events are well represented. However, I have some concerns.

Major concerns

Palpitations is a symptom that may lead to important misleading. It would improve the quality of the paper a better definition of the symptom. For example, a patient whose palpitation is clearly linked to anxiety is not equable to a patient whose palpitation is characterized by sudden onset and termination associated with dizziness. This should be considered a limitation.

The mode of death may be important in this context. Indeed, a death due to the underling infectious disease cannot be considered as a cardiac complication of therapy with azithromycine. If the mode of death cannot be explored, this issue should be reported as a limitation.

Minor concerns

Death usually is not a cause of admission. In the abstract I would move “or death” after “hospitalization”.

The authors in the introduction state that drugs are a common cause of long QT syndrome. I suggest to specify that they are talking about “acquired long QT syndrome”.

Introduction, line 5: missing space before bracket.

It would be clinically helpful to provide a percentage of adverse events for each risk category (low/moderate/high/very high).

Heart failure (HF) did not emerge as a predictor of arrhythmia. As authors state, this is a singular finding, as HF is one of the cardiovascular diseases more strictly linked to arrhythmias. However, I would argue in the discussion section, that HF is an extremely heterogeneous condition. Therefore, some phenotype and some aetiology may be more pro-arrhythmic than others. This issue may explain this singular finding.

In the references number 6,11,14, 23, 24, 25 not all authors are listed, even if in some of these the total number of authors does not exced 6. Please format all the references according to the journal policy (list all the authors if the total amount of authors does not excede 6).

Reviewer #2: The study I read seems particularly interesting to me through its conclusions especially in this period when we face the Covid-19 Pandemic, when many SARSCoV2 patients receive Azithromycin alone or in combination with hydroxychloroquine in the hope of amelioration especially of lung damage. Both drugs are known to have a potential for QT prolongation.

First of all, I want to mention that I am not a statistician but a clinical cardiologist. Even so, I consider that the statistical study made by the authors, imagining two statistical models one for the doctor who prescribes azithromycin and the other for the pharmacist who issues the prescription are very correct. And the conclusions that follow are of great help for both the prescribing doctors and the pharmacists.

The studied population is significant in size and includes all age and sex categories but is an outpatient population for which the data is collected from a database belonging to private insurance companies and therefore does not contain clinical parameters such as history of acute myocardial infarction or heart failure or biological markers such as blood ions that would be of interest in this instance. Nevertheless, the authors mention these aspects as limitations of the study. The current study is retrospective and observational using a database conceived for billing purposes and not for research purposes which represents other limit of the study.

However, doctors can use the score derived from the statistical model in identifying high-risk patients based on clinical conditions preceding the prescription of Azithromycin especially syncope and history of palpitations. Pharmacists can also use the score derived from model that takes into account especially previously prescribed drugs that may prolong the QT interval such as antiarrhythmics, antiemetics, antidepressants or loop diuretics.

For these reasons it is of great practical interest to publish this manuscript provided that the imagined statistical models will be developed through a prospective study.

Reviewer #3: In the paper entitled” Risk of Cardiac Events with Azithromycin – A Prediction Model” by Haridarshan Patel, the authors developed two prediction models, which included baseline demographics, clinical conditions (Model 1), concurrent use of any drug (Model 1) and therapeutic class (Model2) with a risk of QT-prolongation (CQT-Rx) through a splitsample approach, concluded that the Assessment of Cardiac Risk with Azithromycin score maybe helpful to identify patients who are at higher risk of cardiac events following treatment with azithromycin and providers should assess the risk-benefit of using azithromycin and consider alternative antibiotics among high-risk patients.

comments:

1. This is an interesting and fairly well-written paper dealing with identifying patients who are at higher risk of cardiac events following treatment with azithromycin through the Assessment of Cardiac Risk with Azithromycin score.

2. It is well known that ACEIs has protective effect on cardiac remodeling. There is a minor issue that needs attention in order to explain the administration of ACEIs predicts a higher risk of cardiac events following administration with Azithromycin.

6. PLOS authors have the option to publish the peer review history of their article (what does this mean?). If published, this will include your full peer review and any attached files.

Reviewer #1: No

Reviewer #2: No

Reviewer #3: No

---

## [Author Response · Author response to Decision Letter 0]

30 Jul 2020

Please see the attached word doc which includes the responses to both editor and reviewers' feedback.

---

## [Decision Letter · Decision Letter 1]

25 Sep 2020

Risk of Cardiac Events with Azithromycin – A Prediction Model

PONE-D-20-15013R1

Dear Dr. Patel,

We’re pleased to inform you that your manuscript has been judged scientifically suitable for publication and will be formally accepted for publication once it meets all outstanding technical requirements.

Kind regards,

Antonio Cannatà

Academic Editor

PLOS ONE

Additional Editor Comments (optional):

Reviewers' comments:

Reviewer's Responses to Questions

**Comments to the Author**

1. If the authors have adequately addressed your comments raised in a previous round of review and you feel that this manuscript is now acceptable for publication, you may indicate that here to bypass the “Comments to the Author” section, enter your conflict of interest statement in the “Confidential to Editor” section, and submit your "Accept" recommendation.

Reviewer #1: All comments have been addressed

Reviewer #3: All comments have been addressed

2. Is the manuscript technically sound, and do the data support the conclusions?

Reviewer #1: Yes

Reviewer #3: Yes

3. Has the statistical analysis been performed appropriately and rigorously? 

Reviewer #1: Yes

Reviewer #3: No

4. Have the authors made all data underlying the findings in their manuscript fully available?

Reviewer #1: No

Reviewer #3: Yes

5. Is the manuscript presented in an intelligible fashion and written in standard English?

Reviewer #1: Yes

Reviewer #3: Yes

6. Review Comments to the Author

Reviewer #1: Thank you for the revision of the manuscript according to the reviewers' comments. In my opinion the paper has improved its quality.

Reviewer #3: In the paper entitled” Risk of Cardiac Events with Azithromycin – A Prediction Model” by Haridarshan Patel, the authors developed two prediction models, which included baseline demographics, clinical conditions (Model 1), concurrent use of any drug (Model 1) and therapeutic class (Model2) with a risk of QT-prolongation (CQT-Rx) through a splitsample approach, concluded that the Assessment of Cardiac Risk with Azithromycin score maybe helpful to identify patients who are at higher risk of cardiac events following treatment with azithromycin and providers should assess the risk-benefit of using azithromycin and consider alternative antibiotics among high-risk patients.

7. PLOS authors have the option to publish the peer review history of their article (what does this mean?). If published, this will include your full peer review and any attached files.

Reviewer #1: No

Reviewer #3: No

---

## [Editor Report · Acceptance letter]

5 Oct 2020

PONE-D-20-15013R1 

Risk of Cardiac Events with Azithromycin – A Prediction Model 

Dear Dr. Patel:

I'm pleased to inform you that your manuscript has been deemed suitable for publication in PLOS ONE. Congratulations! Your manuscript is now with our production department. 

Kind regards, 

on behalf of

Dr. Antonio Cannatà 

Academic Editor

PLOS ONE